# Isoschaftoside in Fig Leaf Tea Alleviates Nonalcoholic Fatty Liver Disease in Mice via the Regulation of Macrophage Polarity

**DOI:** 10.3390/nu17050757

**Published:** 2025-02-21

**Authors:** Tatsuya Abe

**Affiliations:** Toyo Institute of Food Technology, 23-2, 4-chome, Minami-Hanayashiki, Kawanishi 666-0026, Hyogo, Japan; tatsuya_abe@shokuken.or.jp

**Keywords:** liver disease, NAFLD, macrophage polarity, inflammation, isoschaftoside, *Ficus carica*

## Abstract

Background: Nonalcoholic fatty liver disease (NAFLD) is a subset of fatty liver disease that is not caused by alcohol or viruses, and its increasing incidence presents a major global health concern. As few pharmacotherapies are available for NAFLD, lifestyle modifications, including diet and exercise, serve as the foundation for treatment. Therefore, NAFLD prevention is more important than cure, emphasizing the need for drugs with excellent safety and long-term efficacy. Fig leaf tea contains rutin and isoschaftoside (ISS), which may possess anti-inflammatory properties. Therefore, the aim of this murine-model-based study was to investigate the potential benefits of fig leaf tea in alleviating NAFLD and to determine the underlying mechanism by gene expression analysis. Results: We found that in mice with NAFLD induced by a high-fat diet, the administration of high concentration fig leaf tea or 50 µM ISS significantly ameliorated lobule inflammation. In contrast, low concentration fig leaf tea containing 75 µM ISS did not improve inflammation. The balance between the NAFLD-promoting component of fig leaf tea and the inhibitory effect of ISS was thought to be affected. Gene expression analysis of the liver showed that high concentration fig leaf tea or ISS significantly suppressed the expression of M1 macrophage markers such as CD antigens, toll-like receptors (TLR), chemokines, and cytokines. Further, ISS suppressed the amount of TNF-α released during the M1 polarization of macrophage cells upon lipopolysaccharide (LPS) stimulation. Conclusions: Overall, these results suggest that controlling macrophage polarization may improve NAFLD. Furthermore, these findings highlight the potential clinical applicability of ISS.

## 1. Introduction

Nonalcoholic fatty liver disease (NAFLD) is a symptom of lifestyle-related diseases such as obesity, diabetes, and metabolic syndrome, and can progress similarly to alcoholic liver disease even in the absence of alcohol consumption. The global prevalence of NAFLD was 20% until 2005, but it is currently estimated to be >30%, increasing to 38% in obese children and >57% in obese adults [1]. The highest prevalence of NAFLD is reported in Latin America at 44.4%, followed by the Middle East and North Africa at 36.5% and South Asia at 33.8% [2]. Previously, it was believed that NAFLD comprised two distinct conditions: nonalcoholic fatty liver (NAFL), in which fat is deposited in hepatocytes, and nonalcoholic steatohepatitis (NASH), which poses a risk of progression from fatty liver to cirrhosis or liver cancer because of inflammation and fibrosis. However, in recent years, NAFL and NASH have been regarded as different phases of the same disease [3]. The most important factor in NAFLD development is obesity, which shows a strong positive correlation with the amount of visceral and intrahepatic fat in patients [4,5]. The genotype of the patatin-like phospholipase domain-containing 3 (PNPLA3) gene, which is involved in lipid metabolism, has been identified as a risk factor for NAFLD development [6,7]. The progression of NASH from NAFL is associated with various factors, including oxidative stress, mitochondrial dysfunction, apoptosis, and inflammatory cytokines. Decreased expression of phosphatidylethanolamine-N-methyltransferase (PEMT) in the liver is reported to promote apoptosis and induce hepatitis [8]. Recently, the involvement of autophagy as well as the apoptosis of hepatocytes has been suggested in hepatitis. In mice fed with a high-fat diet for prolonged periods, autophagy is suppressed, promoting hepatitis and ER stress [9].

Kupffer cells, which are macrophages of liver tissues, are also involved in NAFL progression to NASH. Macrophages are central to “innate immunity,” as they detect pathogens or viruses entering the body and break them down through phagocytosis. Macrophages polarize to M1 or M2 phenotypes depending on the type of external stimulus [10]. Macrophages are polarized to type M1 via STAT1, a transcriptional activator, by stimulation with IFN-γ, TNF-α, or LPS, and to type M2 via STAT6 by IL-4 or IL-13 [11,12,13]. M1 macrophages produce inflammatory cytokines such as TNF-α, IL-6, and IL-12 to elicit an inflammatory response, whereas M2 macrophages play a role in wound healing by producing TGF-β and platelet-derived growth factor to reduce excessive tissue injury by M1 [14]. Kupffer cells produce inflammatory factors with an increased M1/M2 ratio as NAFLD progresses [15,16,17]. Tadokoro et al. reported that transplantation of human induced pluripotent stem (iPS) cells derived from fetal liver promotes the polarization of M2 macrophages and suppresses liver fibrosis in mice [18]. Thus, an M2 macrophage-predominant microenvironment is effective in improving NAFLD, but there is insufficient knowledge of the drugs and natural materials that regulate macrophage polarization.

Figs (*Ficus carica* L.) are consumed worldwide in both fresh and processed forms. However, large quantities of leaves and unripe fruits are discarded during fig cultivation. In a previous study, the functional properties of tea made from fig leaves were examined as an unutilized resource [19]. Our clinical study showed that the consumption of fig leaf tea significantly improved skin symptoms in patients with mild atopic dermatitis [20]. Furthermore, blood levels of aspartate aminotransferase (AST) and lactate dehydrogenase (LDH), which are indicators of liver disease, were significantly reduced in patients who consumed fig leaf tea [20]. Animal studies have reported that fig leaf extract has a mitigating effect on drug-induced liver damage [21,22]. Takahashi et al. reported rutin, caffe malate, and isoschaftoside (ISS) as polyphenols present in large amounts in fig leaves [23]. ISS is a c-glycosyl flavonoid found in *Passiflora incarnata* L. and *Abrus cantoniensis* and has been reported to exert several pharmacological effects. Passionflower extract containing ISS has been reported to improve circadian rhythm [24]. Further, the intraperitoneal administration of ISS to mice ameliorated NAFLD during a high-fat diet intake, indicating that ISS has an ameliorative effect on NAFLD [25]. However, the intraperitoneal administration of ISS or other compounds for NAFLD prevention is burdensome for consumers. Therefore, the intake of foods or supplements containing ISS is desirable as a more convenient and sustainable method for NAFLD prevention. In this study, we aimed to determine the ameliorative effect of drinking fig leaf tea or ISS on NAFLD and to elucidate the underlying mechanism through gene expression analysis. The oral consumption of high concentrations of fig leaf tea or ISS improved NASH symptoms. The mechanism has been suggested to be the inhibition of polarized M1 macrophages. The regulation of macrophage polarization in the ISS is the first finding. Apigenin, an ISS aglycon, has been reported to inhibit the progression of osteoarthritis through the suppression of macrophage M1 polarization [26]. The M1 polarization inhibitory effect of ISS is expected to improve liver fibrosis through a different approach to the main reported mechanisms of improvement in liver fibrosis, i.e., the inhibition of fatty acid synthesis and antioxidant effects. The combination of ISS with therapeutic agents with different mechanisms is expected to provide more effective and versatile treatment.

## 2. Materials and Methods

### 2.1. Preparation of Fig Tea

Fig tea was prepared from the leaves of ‘Griśe de Tarascon’ (‘Dauphine’ *sensu* Condit), a fig cultivar that does not contain furanocoumarin [27]. The raw material was collected from our orchard (Kawanishi, Hyogo, Japan). The harvested leaves were washed, cut, steamed at 90 °C for 4 min, and dried at 60 °C for 4 h [19]. An infusion was prepared by steeping tea leaves in water at 80 °C for 3 min. The infusion was freeze dried and then adjusted with water to 0.6° Brix for the low concentration (equivalent to 30 mg/mL, containing 75 µM ISS) and 2.4° Brix for the high concentration (equivalent to 120 mg/mL, containing 270 µM ISS). The low dose fig leaf tea was set at the concentration found to improve atopic dermatitis in human intervention studies [19]. The concentration at the limit for human consumption was defined as the high concentration tea. The ISS content in the fig leaf tea was determined using LC–MS. Analysis was carried out according to the method described by Takahashi et al. [27]. The LC system used was LC-20A (Shimadzu, Kyoto, Japan), the MS system used was micrOTOF Q II (Bruker Daltonics, Billerica, MA, USA), and the column used was Scherzo SM-C18 (Imtakt, Kyoto, Japan). Mobile phase A was water/formic acid (99.7:0.3, *v*/*v*) and B was acetonitrile/formic acid (99:1, *v*/*v*). The gradient began with 0% B, increasing to 100% B at 25 min, and 100% isocratic B from 25 to 35 min. The flow rate of the mobile phase was 0.18 mL/min. The temperature of the column oven was 45 °C. The ISS content of each sample was calculated from the molecular ion peak area of the standard sample (>95% ISS, Cat: 1354, Funakoshi Co., Ltd., Tokyo, Japan).

### 2.2. Animal Study

Four-week-old C57BL/6J male mice were purchased from Japan SLC (Shizuoka, Japan). The mice were kept in individual, well-ventilated cages under a specific-pathogen-free environment at 23 ± 3 °C and 30–70% humidity. The mice were exposed to a 12 h light/dark cycle. The acclimatization period was set to 7 days and body weight was measured on day 7. During the acclimation period, the animals were provided ad libitum access to an ordinary diet, i.e., MF (Oriental Yeast Co., Ltd., Tokyo, Japan) and water. The mice were assigned to five groups using Simple grouping J software ver. 1.0.5 (H&T, Osaka, Japan) to ensure equal mean body weights (Appendix A). During the study period, the samples were orally administered (OA) once a day at 10 mL/kg. The administration period was set at 12 weeks (5–17 weeks of age). The NAFLD-induced group was fed a high-fat diet, i.e., iHFC (Hayashi Kasei CO., Ltd., Osaka, Japan), whereas the non-NAFLD-induced group was fed MF. The components of both feeds are listed in Table 1. Both diets were fed ad libitum, and food intake was measured weekly. The animals were categorized into the following groups: the NF_water group (n = 6), fed MF and OA water; the HF_water group (n = 6), fed iHFC and OA water; the HF_FT low group (n = 6), fed iHFC and OA low concentration fig leaf tea; the HF_FT high group (n = 6), fed iHFC and OA high concentration fig leaf tea; and the HF_ISS group (n = 6), fed iHFC and OA 50 µM ISS. The ISS intake concentration was set at 50 µM, at the lowest level of the fig cultivars, in accordance with Takahashi et al. [27]. The 50 µM ISS sample was prepared by dissolving >95% ISS in water. Body weights were recorded once a week. Blood samples were collected from the tail vein every 2 weeks to measure the blood levels of AST and alanine aminotransferase (ALT). The animal experiments in this study were carried out in accordance with the protocol approved by the Animal Experiment Committee of Toyo Institute of Food Technology (Approval number: 2022-A-001).

### 2.3. Blood Biochemical Test

Blood samples were collected from the posterior vena cava of each mouse at the end of the experiment. The plasma was obtained by centrifugation at 1700× *g* for 10 min at 4 °C. The plasma levels of LDH, LDL cholesterol (LDL-C), triglycerides (TG), cholinesterase (ChE), alkaline phosphatase (ALP), and insulin were measured by Oriental Yeast Co., Ltd.

### 2.4. Histological Analysis

After measuring the liver weight, the lipid droplet area was measured using the anterior right lobe and NAFLD was verified using the medial left lobe. For the lipid droplet area measurement, frozen sections (10 µm) of the liver were prepared, and fat was stained using Oil Red O stain. For NAFLD scoring, paraffin sections (4 µm) were prepared from liver samples fixed in 10% neutral formalin and stained with hematoxylin and eosin (HE). The degrees of steatosis, lobular inflammation, hepatocellular ballooning, and fibrosis in the HE-stained samples were scored according to the NAFLD Activity Score (NAS) system. The NAS scoring system was evaluated semi-quantitatively based on the histologic features of steatosis, lobular inflammation, hepatocellular ballooning, and fibrosis [28]. Histological analysis was contracted to the Kyoto Institute of Nutrition & Pathology, Inc. (Kyoto, Japan).

### 2.5. Gene Expression Analysis

Approximately 0.1 g of the lateral left lobe of the liver was frozen in liquid nitrogen and used for RNA extraction. Total RNA was extracted using the miRNeasy mini kit (Qiagen, Venlo, The Netherlands), and RNA purity was evaluated using an Agilent 2100 Bioanalyzer (Agilent Technologies, Santa Clara, CA, USA). RNA samples with an RNA Integrity Number > 6.7 were used for gene expression analysis. Gene expression analysis was performed using the DNA chip SurePrint G3 Mouse GE 8 × 60 k ver 2.0 (Agilent Technologies). GeneSpring ver. 14.9.1 (Agilent Technologies) was used for extracting the differentially expressed genes and for hierarchical clustering analysis. The quantile method was used to normalize the expression data. Pathway statistical analysis was performed on a pathway collection from the WikiPathways database (2022.05.10) using PathVisio v1.2 (PathVisio 3) tool to determine the pathways showing the most change in expression, considering the number of genes on the pathway measured in the experiment and the number of those that were differentially expressed [29,30].

### 2.6. Cell Culture and Treatment

The RAW 264.7 mouse macrophage cell line was purchased from Cosmo Bio Co., Ltd. (Tokyo, Japan). The cells were cultured in RPMI1640 medium (Thermo Fisher Scientific, Waltham, MA, USA) supplemented with 10% fetal bovine serum (Cosmo Bio Co., Ltd.) under 5% CO_2_ in air at 37 °C. The assessment of cytokine production by LPS stimulation was conducted as reported by He et al. [31], with slight modifications. RAW264.7 cells (1 × 10^6^ cells/mL) were seeded in a 24-well plate (500 µL/well) and cultured for 24 h. After treatment with fig leaf tea or ISS at different concentrations for 6 h, the cells were stimulated with LPS (final 10 ng/mL) for 24 h. The culture supernatants were collected and the concentration of TNF-α was measured using Mouse TNF alpha ELISA Ready-SET-GO! (eBioscience, San Diego, CA, USA). After treatment with fig leaf tea or ISS at different concentrations for 24 h, the cell viability was measured using Cell Counting Kit-8 (MedChemExpress, Monmouth Junction, NJ, USA). All samples except LPS were added at 10% of the medium volume. The concentration of the fig leaf tea added was converted to low concentration fig leaf tea.

### 2.7. Statistical Analysis

Data are expressed as the mean ± standard deviation. Statistical comparisons were carried out using one-way ANOVA. Statistical analyses of the animal and cell culture study were performed using EZR (ver. 2.7.-1), a modified version of R commander designed to include the statistical functions frequently used in biostatistics [32]. GeneSpring ver. 14.9.1 was used for statistical processing in the gene expression analysis. *p* values < 0.05 were considered statistically significant.

## 3. Results

### 3.1. Administration of Fig Leaf Tea and ISS Ameliorate NAFLD in High-Fat-Diet Mice

C57BL/6J mice were fed a high-fat diet ad libitum for 12 weeks to induce NAFLD. During the sample administration period, one mouse each in the HF_FT low and HF_ISS groups showed significant weight loss and was euthanized (HF_FT low: 14 weeks, HF_ISS: 12 weeks). Therefore, these data were excluded from the analysis. The body weights of the NAFLD-induced groups fed iHFC were higher than those of the non-induced group fed MF, and significant differences were observed at 17 weeks.

However, no significant differences were observed between the orally administered samples in the NAFLD group (Figure 1A). The levels of AST and ALT in the blood were also higher in the NAFLD groups than in the non-NAFLD group, confirming that there was a significant difference (*p* > 0.0001) (Figure 1B). However, the levels of both enzymes in the blood were not significantly increased before and after iHFC intake in any of the NAFLD groups, but rather tended to decrease from the initial state. These results indicated that the induction of NAFLD was milder than expected.

The blood levels of LDH, ChE, and LDL-C, which are indicators of NAFLD, were all significantly higher in the NAFLD groups than in the non-NAFLD group, suggesting that fatty liver and hepatitis were induced in the NAFLD groups (Figure 2A–C). The blood levels of TG were significantly higher in the non-NAFLD group than in the NAFLD groups (Figure 2D). iHFC suppresses fat secretion from the liver and induces NAFLD, thus lowering the amount of TG in the blood. These results indicated that fatty liver and hepatitis were induced by iHFC in the NAFLD groups. Meanwhile, no significant differences were observed in any of the test results among the NAFLD groups, and thus no inhibition by the fig leaf tea or by ISS administration could be confirmed. The blood levels of ALP and insulin were not significantly different between all groups (Appendix A).

Liver weights were significantly lower in the non-NAFLD group than in the NAFLD groups (Figure 3A). The NAFLD symptoms in each group were scored using the NAS. HF_water (5.7 ± 0.8) and HF_FT low (5.2 ± 1.3) were considered as indicative of NASH because they exceeded the NASH threshold of 5; the HF_FT high (4.5 ± 0.8) and HF_ISS (4.6 ± 0.9) groups were considered as borderline; and NF_water (0.2 ± 0.4) was considered as non-NASH. The NAS is a system that determines the overall value of steatosis, inflammation, and hepatocellular ballooning. A comparison of the items in each treatment group showed no difference in steatosis but revealed significantly decreased inflammation in HF_FT high and a decreasing trend in HF_ISS (*p* = 0.0576) (Figure 3B–F). Hepatocellular ballooning was not detected in any of the groups. The incidence of fibrosis was 50% (3/6) in HF_water, 20% (1/5) in HF_FT low, and absent in NF_water, HF_FT high, and HF_ ISS (Figure 3D). Fibrosis was observed only around the portal vein. These results indicated that fatty liver was induced by iHFC in HF_FT high and HF_ ISS, but subsequent inflammation and fibrosis were improved. Although the ISS content of the low concentration fig leaf tea was 75 µM and the ISS dose was higher than that of the ISS group (50 µM), lobular inflammation was not improved. This result led us to speculate that fig leaf tea contains a component that inhibits the effects of ISS.

### 3.2. Fig Leaf Tea and ISS Regulate Liver Macrophage Differentiation in NAFLD

The gene expression patterns in the liver of the NF_water, HF_water, HF_FT high, and HF_ISS groups with different hepatitis symptoms were compared. Approximately 50% of the DNA chips used showed expression in the liver. The number of genes expressed in the NF_water group (non NAFLD) was significantly lower than that in the other groups. The gene expression patterns of each individual tended to segregate for each administration sample; however, some individuals remained outliers (Figure 4). Individuals #23 (HF_FT high) and #27 (HF_ISS), classified near HF_water, showed no improvement in inflammation based on histological analysis. Individual #9 (HF_water), classified as HF_FT high, showed no confirmed hepatitis symptoms. These results suggest that the gene expression patterns reflect the symptoms of NAFLD. The number of genes whose expression levels changed more than 2-fold compared with those in NF_water was 2547 in HF_water and approximately 1300 in HF_FT high and HF_ISS (Table 2).

Pathway analysis was performed on 2547 genes that showed altered expression compared with NF_water and HF_water (Table 3 and Table 4). The pathways that included the genes with increased expression were inflammation, oxidative stress, and apoptosis. In contrast, the pathways containing genes with decreased expression were related to the synthesis of fatty acids, cholesterol, and mitochondrial proteins. Furthermore, to extract HF_water-specific altered pathways, pathway analysis was also performed for HF_FT_high and HF_ISS to exclude the pathways that overlapped with those in HF_water. Twenty HF_water-specific activated pathways related to inflammation, cytokines, and chemokines were extracted (Table 3 bold). Interestingly, all pathways for macrophages and microglia registered in the database were specifically activated in HF_water. Pathways that were suppressed in HF_water were not extracted. Genes in the HF_water activated pathways whose expressions were reduced by more than 2-fold in HF_FT high and HF_ISS were extracted (Table 5). The extracted genes were characterized and found to be marker genes of M1 macrophages, which are pro-inflammatory in nature. The inhibition of polarization to M1 macrophages was thus suggested as a mechanism of NASH inhibition by fig leaf tea or ISS administration. Fatty acid synthesis and oxidative stress-related genes involved in progressing NAFLD were not extracted. The expression of genes involved in fibrosis, a step after lobular inflammation, tended to be low (Appendix A).

### 3.3. ISS Inhibits M1 Polarization of Macrophage Cells

The effects of fig tea and ISS on M1 polarization were tested using RAW264.7 murine macrophages. The ISS addition range of 1.6–100 µM did not affect the survival of RAW264.7 cells. However, the viability of these cells was greatly reduced by fig leaf tea at final concentrations >25%; therefore, a 10% concentration was used (Figure 5A). When RAW264.7 was M1 polarized by LPS, ISS addition was found to inhibit the release of TNF-α in a concentration-dependent manner, with significant differences observed above 12.5 µM (Figure 5B). Fig leaf tea at a final concentration of 10% did not inhibit the release of TNF-α, despite the presence of approximately 7.5 µM ISS. These results suggest that the inhibitory components of ISS in fig leaf tea also affect cells, with ISS specifically suppressing M1 macrophage polarization.

## 4. Discussion

The effect of fig tea on NAFLD was investigated, focusing on its ability to improve liver function. NAFLD symptoms such as steatosis, lobular inflammation, and fibrosis were observed in mice fed a high-fat diet of iHFC. The induction of NAFLD symptoms by iHFC was not improved by the ingestion of the low concentration fig leaf tea. In contrast, steatosis was induced in the high concentration fig leaf tea and ISS groups, but subsequent lobular inflammation and fibrosis were ameliorated. The ameliorative effect of ISS on hepatitis was observed at 50 µM, whereas it was not observed at the low concentration of fig leaf tea, which contained approximately 75 µM of ISS. These results suggest that ISS exerts an ameliorative effect on lobular inflammation and fibrosis, and that fig leaf tea contains components that promote these symptoms or inhibit the effects of ISS. The fig cultivar in this study contains no furanocoumarin, but about 16 µM of the furanocoumarin glycoside psoralenoside (PO). Psoralen, an aglycon of PO, has been reported to have anti-inflammatory effects in ligament cells, while it has also been reported to activate endoplasmic reticulum stress by increasing PERK and ATF6 gene expression in the liver cell line HepG2 [33,34]. Acute toxicity studies of psoralen in C57BL6 mice have also reported that it causes liver damage [35]. It has been suggested that HF-induced NAFLD may be accelerated if PO is metabolized to psoralen. The reason for the failure of low doses of fig tea to suppress lobular inflammation is thought to be that the inhibitory effect of ISS was offset by the NAFLD-promoting effect of psoralen. Su et al. reported that the intraperitoneal administration of ISS to mice with palmitic acid (PA)-induced NAFLD improved steatosis by inhibiting the PA-induced autophagy suppression [25]. As the administration method and ISS concentration differed from those used in this study, the two studies cannot be compared. However, these results suggest that ISS directly reaches the liver and exerts physiological activity. The ISS inhibitory effect of fig leaf tea may involve absorption and metabolism in the intestinal tract. It has been reported that C-glycosyl flavonoids, to which ISS belongs, are metabolized at different rates depending on the intestinal flora [36]. Therefore, it was considered possible that differences in the intestinal flora could also influence the metabolic uptake of ISS.

In this study, iHFC was used to induce NAFLD. In similar studies, methionine- and choline-depleted diets (MCDs) have been used as induction diets. MCDs have been mainly used for NAFLD induction. However, loss of muscle mass and body weight has been reported with MCD consumption, making it unsuitable for evaluating continuous improvement [37]. The cholic acid in iHFC inhibits lipid elimination from the liver [38]. Furthermore, due to its high lipid content, iHFC is expected to promote fatty acid synthesis by cholesterol, rendering this experimental system proficient in inducing fatty liver under near-ordinary conditions. In this study, iHFC consumption-induced steatosis, lobular inflammation, and fibrosis in liver cells and could be evaluated using the NAS system. However, no increase was observed in AST or ALT levels, which are commonly associated with hepatitis. Viral hepatitis characterized by extensive cell death is associated with a marked increase in AST and ALT levels, whereas NAFLD does not typically result in substantial cell death, and thus, ALT and AST levels are not elevated in some cases. Therefore, the absence of dead cells in this study was considered to have led to a lack of an increase in the levels of both of the enzymes. As this study focused on the preventative properties of foods in relation to NAFLD rather than treatment, the combination of C57BL/6J and iHFC, which does not show any notable signs of disease, was used. The use of TSOD and TSNO mice, which show severe symptoms of NAFLD in combination with iHFC, may demonstrate a more definite effect of ISS [38]. Gene expression analysis of the liver showed a significant increase in the number of genes expressed in the NAFLD groups compared with those in the non-NAFLD group. These results may thus reflect a group of genes expressed in NAFLD development. Pathway analysis showed that the genes involved in inflammation and oxidative stress were activated in the NAFLD groups compared with those in the non-NAFLD group, suggesting the role of these pathways in the development of steatosis and hepatitis. The suppressed pathways included those related to fatty acid and cholesterol synthesis. A negative feedback effect was thus speculated for excessive lipid accumulation by iHFC. In mice treated with high concentration fig leaf tea and ISS, gene expressions characteristic of pro-inflammatory M1 macrophages were suppressed. Furthermore, the gene expressions of inflammatory cytokines and chemokines produced by M1 macrophages were also decreased. These results suggest that the hepatitis-ameliorating effect of ISS is via the inhibition of M1 macrophage polarization. Thus, the suppression of M1 macrophage polarization may have resulted in the improved hepatitis symptoms in the high concentration fig leaf tea and ISS-treated groups. Macrophages polarize to the M1 or M2 type depending on the external stimulus. Eleven species of Toll-like receptors (TLRs) have been reported in mammals, and a specific receptor responds to a specific type of stimulus [39]. TLR4 recognizes lipopolysaccharide (LPS) from Gram-negative bacteria whereas TLR2 recognizes peptidoglycan from Gram-positive bacteria. When TLR4 and TLR2 recognize their ligand molecules, they induce the production of inflammatory cytokines and chemokines via the transcription factor NF-κB. Mice fed a high-fat diet showed an increased production of inflammatory cytokines in the Kupffer cells and increased hepatitis, but these effects were suppressed in TLR4-knockout mice, and blood ALT levels were reduced [40]. In the present study, the HF_ISS group showed a significant decrease in TLR2 and TLR4 gene expression, suggesting that a decrease in LPS sensitivity reduced the polarization of M1 macrophages and improved NASH. In RAW264.7 cells, ISS suppressed the LPS-stimulated induction of M1 polarization, supporting the results of the gene expression analysis (Figure 5). Further, a study using BV2 microglia, i.e., brain-resident macrophages, reported that the induction of inflammatory cytokines and reactive oxygen by LPS is suppressed by treatment with ISS [41]. This report suggests that ISS functions directly on macrophages and supports our postulated mechanism.

The HF_FT high and HF_ISS groups showed a decreased expression of several chemokines and chemokine receptors. Chemokines are broadly classified as “inflammatory chemokines” that function in acute and chronic inflammation and “homeostatic chemokines” that function in homeostasis [42]. In the present study, the gene expression of inflammatory chemokines such as CCL2, CCL3, CCL4, and CCL5 was decreased. In particular, CCL2 and its receptor, CCR2, promote colon inflammation and tumorigenesis by inducing the migration of monocytes and macrophages [43,44]. CCL2 and CCL5 expression are increased in both human and mouse NASH and are closely associated with NASH development [45,46,47]. Cencriviroc, an antagonist of CCL2 and CCL5 receptors (CCR2/CCR5), is reportedly beneficial, with notably improved liver fibrosis in the treated patients [48]. Deletion of the CCL3 gene in mice resulted in a decrease in the M1/M2 ratio and showed improvement in hepatitis when the mice were on a high-fat diet [49]. Overall, these results suggest that ISS induces a reduction in CCL2, CCL3, CCL5, and CCR2 gene expression and contributes to NASH improvement. Gene expression of the chemokine receptor CX3CR1 was also found to decrease; CX3CR1 and its ligand CX3CL1 have been reported to contribute to atopic dermatitis [50].

This study is valuable in showing that orally administered high concentration fig leaf tea and its component ISS ameliorate high-fat diet-induced NAFLD via macrophages. The limitation of this study was that the inhibitory effect of ISS on macrophage M1 polarization was not confirmed at the protein level. Gene expression analysis showed that not only were inflammatory cytokine genes produced by M1 macrophages suppressed, but also CD antigens and the gene expression of TRLs involved in M1 polarization. These data suggested that ISS action was not part of M1 macrophage function but involved more in upstream polarization. In the future, immunostaining and flow cytometry analyses using M1/M2 marker antibodies are expected to reveal the regulatory function of ISS in macrophage polarization.

## 5. Conclusions

This study investigated the effects of fig leaf tea and its component ISS on NAFLD. The results showed that a high concentration of fig leaf tea and ISS ameliorated NASH symptoms. However, fig leaf tea at low concentrations did not improve the condition. The balance between the NAFLD-promoting component of the tea and the inhibitory effect of ISS was thought to be affected. Gene expression analysis revealed the suppressed expression of many M1 macrophage markers, including CD antigens, TLRs, and inflammatory cytokines. The results of the gene expression analysis suggested that the improvement of NASH in ISS occurred via the suppression of M1 macrophage polarization. A long-term administration of ISS will show its durability and safety. It is hoped that ISS will become a preventive measure against NAFLD through human intervention trials using these findings.

## Figures and Tables

**Figure 1 nutrients-17-00757-f001:**
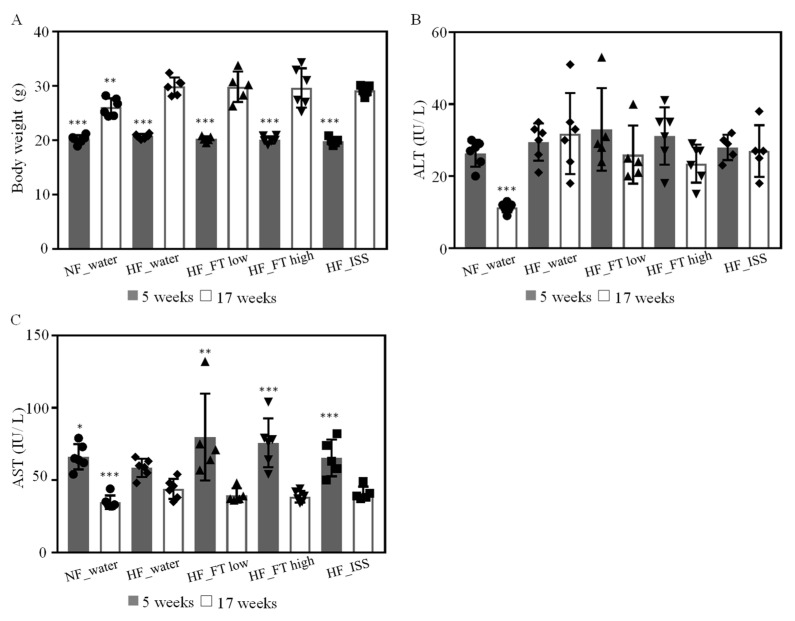
Effects of fig tea and ISS on body weight and blood ALT and AST levels. (**A**) Body weight, (**B**) ALT, and (**C**) AST before and after NAFLD induction. Blood ALT and AST levels were measured from the tail vein at 5 weeks and from the vena cava at 17 weeks. The results are the mean ± SD, with n ≥ 5 per group. * *p* < 0.05, ** *p* < 0.01, *** *p* < 0.001 compared with HF_water.

**Figure 2 nutrients-17-00757-f002:**
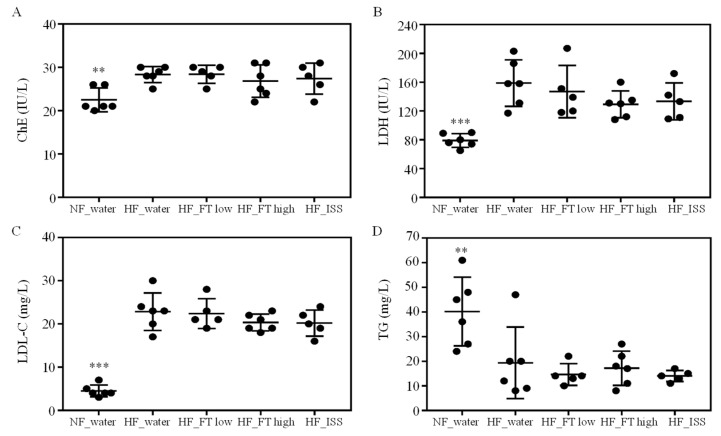
Effects of fig tea and ISS on blood parameters. Blood levels of (**A**) cholinesterase (ChE), (**B**) lactate dehydrogenase (LDH), (**C**) low-density lipoprotein (LDL) cholesterol, and (**D**) triglycerides (TG) at 17 weeks. Blood samples were taken from the vena cava. The results are the mean ± SD, with n ≥ 5 per group. ** *p* < 0.01, *** *p* < 0.001 compared with HF_water.

**Figure 3 nutrients-17-00757-f003:**
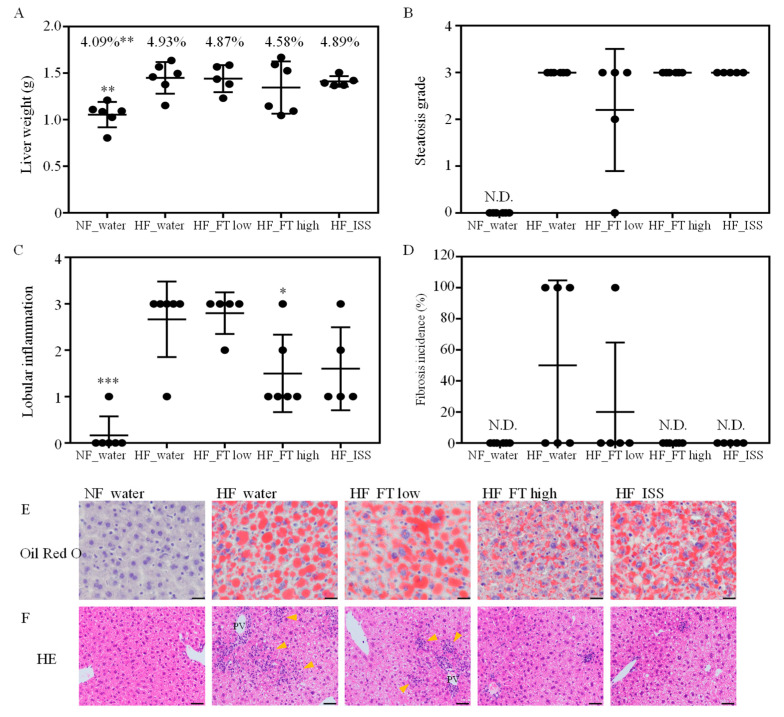
Effects of fig tea and ISS on NAFLD symptoms. (**A**) Liver weight, % values are relative to body weight. (**B**) Steatosis grade. (**C**) Lobular inflammation, scored based on the NAS system. (**D**) Fibrosis incidence. The results are the mean ± SD, with n ≥5 per group. * *p* < 0.05, ** *p* < 0.01, *** *p* < 0.001 compared with HF_water. (**E**) Oil Red O staining of liver tissue. Scale bar, 20 µm. Lipid deposition (red staining) was observed in all NAFLD-induced groups. (**F**) Hematoxylin and eosin (HE) staining of liver tissue. Scale bar, 10 µm. Inflammation areas show accumulation of lymphocytes (arrow heads). The accumulation of many inflammatory cells was observed around the portal vein (PV) in the HF_water and HF_FT low groups.

**Figure 4 nutrients-17-00757-f004:**
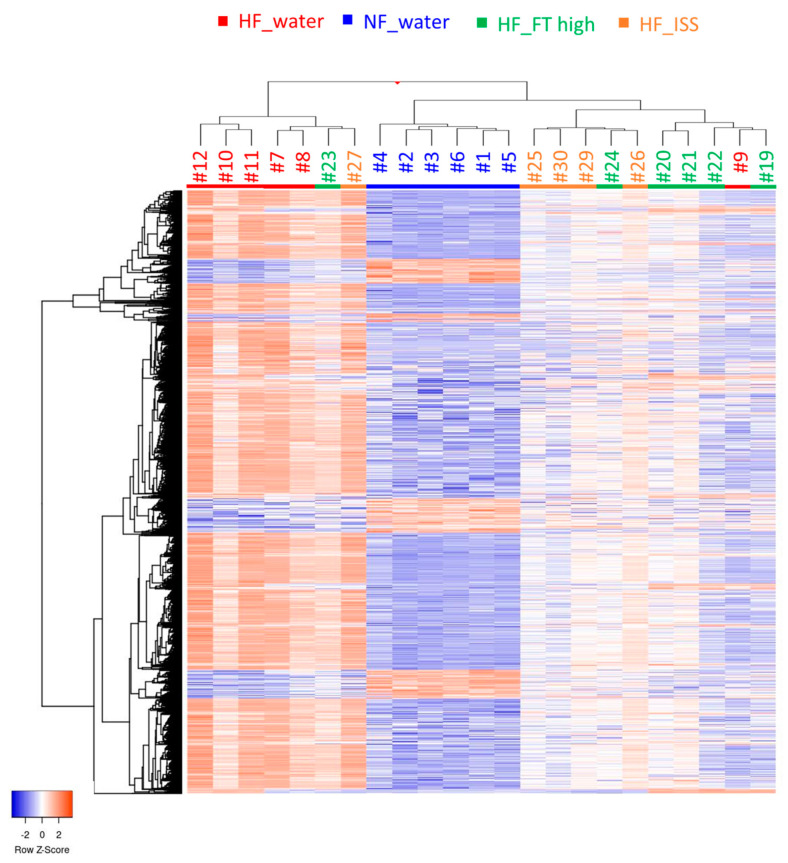
Heatmap of the differentially expressed genes. Red and blue represent upregulated and downregulated genes, respectively. White represents non-significant changes in gene expression. #numbers indicate individual mice.

**Figure 5 nutrients-17-00757-f005:**
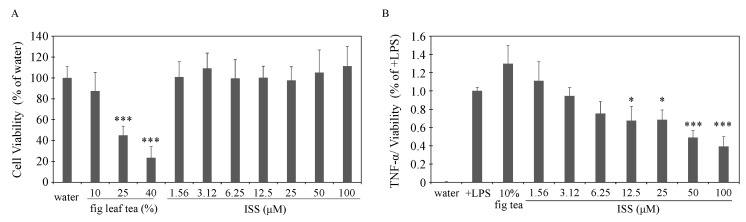
Inhibitory effect of ISS on the M1 polarization of macrophages. (**A**) Viability of RAW264.7 cells at 24 h after treatment with different concentrations of fig leaf tea and ISS. *** *p* < 0.001 compared with water. (**B**) The effect of fig leaf tea and different concentrations of ISS on TNF-α release by LPS. * *p* < 0.05, *** *p* < 0.001 compared with +LPS.

**Table 1 nutrients-17-00757-t001:** Constituents of the administered feeds.

	MF	iHFC
Nutrients (g/100 g)		
Protein	23.1	16.1
Fat	5.1	31.0
Ash	5.8	4
Fiber	2.8	1.9
Carbohydrate	55.3	38.4
Moisture	7.9	5.5
Cholesterol	0	2.5
Sodium cholate	0	0.5
Energy (kcal/100 g)	359.5	497.4
Fat–energy ratio (%)	12.8	56.1

**Table 2 nutrients-17-00757-t002:** Overview of the microarray analysis.

	NF_Water	HF_Water	HF_FT High	HF_ISS
Inflammation	none	severe	mild	mild
Samples	6	6	6	5
Detection probes	29,960.8 (52.8%)	32,618.3 ^†^ (57.5%)	32,285.3 ^†^ (56.9%)	31,520.2 ^†^ (55.5%)
Incremental probes	―	2149	1009	1145
Decline probes	―	398	286	241

The number of increased and decreased genes showed a significant difference at *p* < 0.05 compared to those in the NE_water group, indicating a more than 2-fold variation in expression. ^†^ *p* < 0.05 compared with NF_water.

**Table 3 nutrients-17-00757-t003:** Pathways containing activated genes in the HF_water group.

Pathway (Activated)	ID	Genes Found in Pathways [n]	NF_Water		Fig-High		ISS		Listed Number
			Genes Meeting Criterion [r]	Z Score	Genes Meeting Criterion [r]	Z Score	Genes Meeting Criterion [r]	Z Score	
**Microglia pathogen phagocytosis pathway**	WP3626	41	26	10.7	26	10.97	22	10.03	1
**Tyrobp causal network in microglia**	WP3625	58	31	10.31	30	10.17	18	5.9	2
**Type II interferon signaling (IFNG)**	WP1253	33	21	9.63	21	9.88	17	8.56	3
**Chemokine signaling pathway**	WP2292	177	54	8.35	53	8.43	44	7.49	4
**Macrophage markers**	WP2271	10	8	6.94	8	7.1	7	6.75	5
**Inflammatory response pathway**	WP458	30	13	5.63	14	6.4	9	4.03	6
**Spinal cord injury**	WP2432	99	27	5.16	29	6.05	25	5.7	7
**IL-5 signaling pathway**	WP151	69	21	5.14	21	5.35	14	3.3	8
**Oxidative stress and redox pathway**	WP4466	91	25	5.01	20	3.52	17	3.26	9
**Lung fibrosis**	WP3632	60	18	4.68	21	6.13	16	4.81	10
**Burn wound healing**	WP5056	21	9	4.64	8	4.08	8	4.67	11
Kit receptor signaling pathway	WP407	67	19	4.52	20	5.11	10	1.71	12
**Circulating monocytes and cardiac macrophages in diastolic dysfunction**	WP4474	4	3	4.07	3	4.17	3	4.61	13
**Cytokines and inflammatory response**	WP222	29	10	4.02	10	4.16	8	3.51	14
**Fibrin complement receptor 3 signaling pathway**	WP5128	44	13	3.91	13	4.07	13	4.78	15
Oxidative damage response	WP1496	41	12	3.71	8	1.84	6	1.27	16
**ApoE and miR-146 in inflammation and atherosclerosis**	WP3592	8	4	3.5	3	2.46	3	2.82	17
**Apoptosis**	WP1254	78	18	3.39	18	3.57	13	2.38	18
**IL-3 signaling pathway**	WP373	99	21	3.23	24	4.41	17	2.87	19
Estrogen metabolism	WP1264	9	4	3.18	1	0.04	0	0.94	20
Eicosanoid lipid synthesis map	WP4335	9	4	3.18	2	0.86	1	1.47	21
**Prostaglandin synthesis and regulation**	WP374	31	9	3.18	9	3.32	7	2.65	22
Eicosanoid metabolism via cyclooxygenases (COX)	WP4347	32	9	3.07	2	1.73	2	1.13	23
Myometrial relaxation and contraction pathways	WP385	149	28	3.02	24	2.18	19	1.62	24
**Osteoclast signaling**	WP454	14	5	2.93	4	2.17	4	2.56	25
Eicosanoid synthesis	WP318	19	6	2.84	3	0.72	3	1.04	26
**IL-2 signaling pathway**	WP450	76	16	2.78	17	3.32	13	2.49	27

Pathways indicated in bold represent the HF_water group-specific changes. n: total number of genes in the pathway, r: number of altered genes (*p* < 0.05) in the pathway.

**Table 4 nutrients-17-00757-t004:** Pathways containing suppressed genes in the HF_water group.

Pathway (Suppressed)	ID	Genes Found in Pathways [n]	NF-Water		HF_FT High		HF_ISS	
			Genes Meeting Criterion [r]	Z Score	Genes Meeting Criterion [r]	Z Score	Genes Meeting Criterion [r]	Z Score
Cholesterol biosynthesis	WP103	15	14	23.87	0	0.55	0	0.67
Cholesterol metabolism with Bloch and Kandutsch–Russell pathways	WP4346	56	23	19.73	6	4.7	2	0.30
Regulation of Pgc1a expression by a Gsk3b-Tfeb signaling axis in the skeletal muscle	WP4763	4	2	6.46	0	0.28	0	0.35
Fatty acid biosynthesis	WP336	22	4	5.06	1	0.86	2	1.73
Omega-9 fatty acid synthesis	WP4351	14	3	4.86	3	5.22	1	0.94
Omega-3/omega-6 fatty acid synthesis	WP4350	15	3	4.65	3	5.00	1	0.87
Amino acid conjugation of benzoic acid	WP1252	2	1	4.56	0	0.20	0	0.24
Oxidation by cytochrome P450	WP1274	40	5	4.4	2	1.37	2	0.79
Selenium micronutrient network	WP1272	28	4	4.32	3	3.32	2	1.34
SREBF and miR33 in cholesterol and lipid homeostasis	WP2084	11	2	3.58	0	0.47	0	0.57
Folic acid network	WP1273	25	3	3.3	2	2.16	1	0.33
Steroid biosynthesis	WP55	13	2	3.21	0	0.51	0	0.62
PPAR signaling pathway	WP2316	80	6	3.2	7	4.37	4	1.12
Alanine and aspartate metabolism	WP240	15	2	2.91	0	0.55	1	0.87
Eicosanoid metabolism via cytochrome P450 mono-oxygenases	WP4349	16	2	2.78	1	1.22	1	0.80
Statin pathway	WP1	19	2	2.44	0	0.62	0	0.76
Metapathway biotransformation	WP1251	136	7	2.32	4	0.81	6	1.06
ESC pluripotency pathways	WP339	117	6	2.13	5	1.79	6	1.44
G protein signaling pathways	WP232	91	5	2.11	0	1.37	1	1.04
Amino acid metabolism	WP662	95	5	2.01	1	0.66	4	0.76
Glycolysis and gluconeogenesis	WP157	49	3	1.84	1	0.03	2	0.49

n: total number of genes in the pathway, r: number of altered genes (*p* < 0.05) in the pathway.

**Table 5 nutrients-17-00757-t005:** Genes upregulated in the HF_water group.

Gene ID	Gene Symbol	Gene Name	Relative Expression (vs. NF_Water)			Relative Expression (vs. HF_Water)		Listed Number in Table 3
			HF_Water	HF_FT High	HF_ISS	HF_FT High	HF_ISS	
16452	*Jak2*	Janus kinase 2	1991.6	22.5	4.1	0.01	>0.01	3, 4, 5, 10, 19, 27
19225	* **Ptgs2** *	**prostaglandin-endoperoxide synthase 2**	6.9	1.3	2.0	0.19	0.29	22
16364	*Irf4*	interferon regulatory factor 4	10.9	2.1	2.7	0.19	0.25	3, 18
20306	*Ccl7*	chemokine (C-C motif) ligand 7	14.3	3.3	4.3	0.23	0.30	4
12842	*Col1a1*	collagen, type I, alpha 1	7.2	1.7	3.0	0.23	0.42	6, 25
20311	* **Cxcl5** *	**chemokine (C-X-C motif) ligand 5**	6.7	1.6	3.0	0.24	0.45	4
20304	* **Ccl5** *	**chemokine (C-C motif) ligand 5**	21.9	5.5	11.5	0.25	0.52	4, 10
12475	*Cd14*	CD14 antigen	8.7	2.2	3.3	0.26	0.38	5, 15
21926	* **Tnf** *	**tumor necrosis factor**	22.8	5.9	9.2	0.26	0.40	7, 9, 10, 11,14, 15, 18
12772	*Ccr2*	chemokine (C-C motif) receptor 2	20.3	5.2	6.7	0.26	0.33	4, 8, 10, 13
15978	* **Ifng** *	**interferon gamma**	22.1	6.1	7.2	0.27	0.32	3, 6, 7, 14
20296	* **Ccl2** *	**chemokine (C-C motif) ligand 2**	16.1	4.4	7.0	0.28	0.43	4, 7, 10, 11, 13, 15
17329	*Cxcl9*	chemokine (C-X-C motif) ligand 9	22.7	6.3	11.6	0.28	0.51	3, 4
20302	* **Ccl3** *	**chemokine (C-C motif) ligand 3**	12.1	3.6	5.5	0.30	0.46	4, 10
12843	*Col1a2*	collagen, type I, alpha 2	3.8	1.1	1.7	0.30	0.45	6
16176	* **Il1b** *	**interleukin 1 beta**	9.3	2.9	3.3	0.31	0.36	3, 7, 10, 11, 14
15945	* **Cxcl10** *	**chemokine (C-X-C motif) ligand 10**	13.1	4.1	6.2	0.31	0.47	3, 4, 7, 15
20310	* **Cxcl2** *	**chemokine (C-X-C motif) ligand 2**	2.7	0.9	1.1	0.31	0.40	4, 7, 10
20303	* **Ccl4** *	**chemokine (C-C motif) ligand 4**	22.0	7.4	9.8	0.34	0.44	4, 10
20307	* **Ccl8** *	**chemokine (C-C motif) ligand 8**	23.9	8.0	11.9	0.34	0.50	4
13051	*Cx3cr1*	chemokine (C-X3-C motif) receptor 1	3.9	1.3	1.6	0.35	0.42	4
24088	*Tlr2*	toll-like receptor 2	12.6	4.5	6.1	0.36	0.48	17
15979	* **Ifngr1** *	**interferon gamma receptor 1**	25.8	9.5	14.7	0.37	0.57	3
20305	*Ccl6*	chemokine (C-C motif) ligand 6	6.1	2.3	3.0	0.38	0.49	4
12524	* **Cd86** *	**CD86 antigen**	4.4	1.7	2.2	0.39	0.50	5, 6
20375	*Spi1*	spleen focus forming virus (SFFV) proviral integration oncogene	6.2	2.7	2.8	0.43	0.44	3, 17, 19
12825	*Col3a1*	collagen, type III, alpha 1	2.8	1.2	1.0	0.43	0.37	6
16193	* **Il6** *	**interleukin 6**	3.7	1.6	1.0	0.44	0.26	7, 10, 11, 14, 15
12519	* **Cd80** *	**CD80 antigen**	2.3	1.2	1.0	0.51	0.42	6
21898	*Tlr4*	toll-like receptor 4	4.7	3.1	2.3	0.65	0.50	7, 15, 17

Genes marked in bold have antibodies marketed as M1 macrophage markers (https://www.funakoshi.co.jp/contents/63980, accessed on 23 April 2023). Listed number: pathways listed in Table 3 including these genes.

## Data Availability

The data used to support the findings of this study are available from the corresponding author upon reasonable request.

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
