# Peer review of "Isoschaftoside in Fig Leaf Tea Alleviates Nonalcoholic Fatty Liver Disease in Mice via the Regulation of Macrophage Polarity"

_nutrients, 2025, doi:10.3390/nu17050757_

Round 1
Reviewer 1 Report
Comments and Suggestions for Authors
Tatsuya Abe has investigated the beneficial effects of fig tea against NAFLD development in mice. The findings of this animal study are interesting, but some concerns should be addressed:
Major points:
- What rationale was used for fig tea's high and low concentration and 50 µM ISS doses?
- For liver fibrosis analysis, Sirius Red or Masson's trichrome staining should be used rather than hematoxylin-eosin staining. This reviewer cannot observe fibrosis in Figure 3 nor in the supplementary material file (photomicrographs).
- How does the author explain that serum ALT and AST levels presented higher values at 5 weeks than at 17 weeks? (see figure 1)
- Raw data from the gene expression analysis should be presented as supplementary material.
- In the legend of Figure 3. Use “Inflammation areas show accumulation of lymphocytes (include arrows) rather than “blue granules”. Use “Accumulation of many inflammatory cells were observed around the portal vein (PV) in the HF_water and HF_FT low groups.
Minor points:
- Use lipid droplets area but not adipocyte area
- Were the animals 12h-fasted for serum biochemical analysis?
This work has interesting findings to be published in Nutrients.
Author Response
Dear Reviewer 1
I wish to express my appreciation to the Reviewer for your insightful comments, which have helped me significantly improve the paper.
Tatsuya Abe has investigated the beneficial effects of fig tea against NAFLD development in mice. The findings of this animal study are interesting, but some concerns should be addressed:
Response: I thank the Reviewer for this pertinent comment. I have addressed the points you raised as follows. I hope these answers meet your expectations.
Major points:
- What rationale was used for fig tea's high and low concentration and 50 µM ISS doses?
Response: The concentration of ISS was 50 µM, which was the lowest level of the fig cultivars investigated in the previous study. Please check the additional description (section 2.2.).
- For liver fibrosis analysis, Sirius Red or Masson's trichrome staining should be used rather than hematoxylin-eosin staining. This reviewer cannot observe fibrosis in Figure 3 nor in the supplementary material file (photomicrographs).
Response: I believe that your point is correct. Unfortunately, it was difficult for me to carry out the historical analysis due to my lack of expertise. So I contacted a specialist company, Kyoto Institute of Nutrition & Pathology, Inc. They told me that HA staining could be used to identify them, so I used it. Future follow-up experiments will consider implementing sirius red staining. Gene expression analysis shows that the group of genes associated with fibrosis is reduced in HF_FT high and HF_ISS, suggesting that fibrosis is suppressed. Please check the supplementary Data S3.
- How does the author explain that serum ALT and AST levels presented higher values at 5 weeks than at 17 weeks? (see figure 1)
Response: As pointed out, ALT and AST are well below the average values for 17 weeks old Bl6 mice. On the other hand, food intake, body weight and liver weight were not significantly different from the average. Health observations also showed no abnormalities, so the study was continued.
- Raw data from the gene expression analysis should be presented as supplementary material.
Response: As noted, the raw data was presented.
- In the legend of Figure 3. Use “Inflammation areas show accumulation of lymphocytes (include arrows) rather than “blue granules”. Use “Accumulation of many inflammatory cells were observed around the portal vein (PV) in the HF_water and HF_FT low groups.
Response: Thank you for your suggestion, it has been amended.
Minor points:
- Use lipid droplets area but not adipocyte area
Response: As you mentioned, we have corrected this.
- Were the animals 12h-fasted for serum biochemical analysis?
Response: Blood samples were also taken without fasting, as AST and ALT measurements over time did not include fasting.
Sincerely,
Tatsuya Abe
Reviewer 2 Report
Comments and Suggestions for Authors
- In the manuscript “Isoschaftoside in fig leaf tea alleviates nonalcoholic fatty liver disease in mice via the regulation of macrophage polarity”, a fig leaf tea was used as a potential natural therapy to alleviate NAFLD in mice, demonstrating that modulation of macrophage functions and inflammation may be key mechanisms. Here are some suggested improvements:
- Include the initial body weight of C57BL/6J male mice in section 2.2.
- Consider investigating other parameters of liver function, such as alkaline phosphatase (ALP), and blood parameters like glucose and albumin levels.
- Was water intake measured in the in vivo study?
- Provide more detail on the rationale for the doses and concentrations used, and include appropriate references.
- Was the fig leaf tea extract directly diluted into the cell culture medium?
- Gene expression pathways should be explored in the in vitro experiments.
- Based on the current in vitro results, can we conclude that fig leaf tea suppressed the M1 macrophage polarization of RAW264.7 cells after LPS challenge?
- Ensure consistency in the use of abbreviations throughout the text.
Author Response
Dear Reviewer 2
I wish to express my appreciation to the Reviewer for your insightful comments, which have helped me significantly improve the paper.
- In the manuscript “Isoschaftoside in fig leaf tea alleviates nonalcoholic fatty liver disease in mice via the regulation of macrophage polarity”, a fig leaf tea was used as a potential natural therapy to alleviate NAFLD in mice, demonstrating that modulation of macrophage functions and inflammation may be key mechanisms. Here are some suggested improvements:
Response: I thank the Reviewer for this pertinent comment. I have addressed the points you raised as follows. I hope these answers meet your expectations.
- Include the initial body weight of C57BL/6J male mice in section 2.2.
Response: Supplementary data on body weight when grouped. Please check the supplementary Data S1.
- Consider investigating other parameters of liver function, such as alkaline phosphatase (ALP), and blood parameters like glucose and albumin levels.
Response: The blood level of Insulin and ALP were presented in the supplement data as parameters related to liver function and NAFLD. Please check the supplementary Data S3. Glucose level was also measured, but this was not presented because the animals were not fasted for the convenience of the experiment.
- Was water intake measured in the in vivo study?
Response: It was not measured in this study. However, the sample was administered orally by catheter, so we believe that the prescribed dose (10 mL/g) was given.
- Provide more detail on the rationale for the doses and concentrations used, and include appropriate references.
Response: As noted, the rationale for the dosing concentrations has been provided. Please check the section 2.1. and 2.2..
- Was the fig leaf tea extract directly diluted into the cell culture medium?
Response: In this experiment, both tea and ISS solutions were diluted with water and the amount added was fixed at 10% of the culture medium. Samples with 10% of the culture medium added with water are used as Control.
- Gene expression pathways should be explored in the in vitro experiments.
- Based on the current in vitro results, can we conclude that fig leaf tea suppressed the M1 macrophage polarization of RAW264.7 cells after LPS challenge?
Response to both comments: The suppression of M1 polarisation in ISS is a hypothesis inferred from gene expression analysis in the liver. Initially, TNF-α release was measured after the addition of LPS to confirm that the ISS also works in vitro. The results confirmed a concentration-dependent inhibitory effect of ISS, providing some support for the hypothesis. Regarding the gene expression you mentioned, we think it is very important and interesting. We will implement it in the future.
- Ensure consistency in the use of abbreviations throughout the text.
Response: Thank you for pointing this out, it has been amended.
Sincerely,
Tatsuya Abe
Reviewer 3 Report
Comments and Suggestions for Authors
Abstract
The abstract provides a good overview but lacks specific limitations of the study and potential weaknesses in the experimental design.
The results regarding the effects of fig leaf tea on NAFLD are somewhat general and should be more precisely summarized.
Introduction
While the introduction explains NAFLD pathophysiology in detail, the link between isoschaftoside (ISS) and macrophage polarization is not clearly established.
More emphasis is needed on why ISS is a particularly promising therapeutic approach compared to existing mechanisms.
The epidemiological context is addressed, but more detailed data on incidence trends in different populations would add value.
Materials and Methods
The methodology for ISS dosing and fig leaf tea preparation is described, but it remains unclear whether the administered doses are directly translatable to humans.
More details on statistical analysis are needed, particularly regarding the control of potential confounders.
The sample size (n = 6 per group) is relatively small, which may limit the statistical power of the findings.
No discussion on potential dietary variations or microbiome influences that could affect the results.
Results
While significant reductions in inflammatory markers were observed in the HF_FT_high group, it is unclear why the HF_FT_low group did not show similar effects. A more detailed explanation is needed.
The differences between ISS alone and fig leaf tea are mentioned, but no in-depth analysis is provided on potential antagonistic or synergistic compounds in fig leaf tea.
The gene expression data are extensive, but their functional relevance to known NAFLD pathogenesis mechanisms is not sufficiently discussed.
No clarification on whether the observed inhibition of M1 polarization persists long-term.
Discussion
The study proposes a hypothesis about the role of M1/M2 macrophage polarization in NAFLD but does not compare it with other established mechanisms of NAFLD progression.
The claim that ISS inhibits TLR4-mediated inflammatory responses lacks direct functional experiments (e.g., knockdown or knockout models) to confirm the mechanism.
While clinical translation is briefly mentioned, potential safety concerns or side effects of long-term ISS supplementation are not addressed.
No discussion on possible sex-specific differences or individual variability in response to ISS or fig leaf tea.
Conclusion
The conclusion is broad and somewhat speculative, implying that ISS has general anti-inflammatory properties without clearly limiting its effects to NAFLD.
More precise suggestions for future research are needed, particularly regarding clinical trials or alternative models to investigate the long-term effects of ISS.
Overall Assessment
The study presents interesting findings on ISS in NAFLD progression but has gaps in methodology and mechanistic explanations. The unclear distinction between fig leaf tea and ISS effects, as well as the lack of consideration for potential side effects and confounding factors, should be addressed.
Author Response
Dear Reviewer 3
I wish to express my appreciation to the Reviewer for your insightful comments, which have helped me significantly improve the paper.
Overall Assessment
The study presents interesting findings on ISS in NAFLD progression but has gaps in methodology and mechanistic explanations. The unclear distinction between fig leaf tea and ISS effects, as well as the lack of consideration for potential side effects and confounding factors, should be addressed.
Response: I thank the Reviewer for this pertinent comment. As suggested, I have increased the explanation of the methodology and mechanism. I have also described the differences between fig leaf tea and ISS based on previous information. I hope these answers meet your expectations.
Abstract
The abstract provides a good overview but lacks specific limitations of the study and potential weaknesses in the experimental design.
The results regarding the effects of fig leaf tea on NAFLD are somewhat general and should be more precisely summarized.
Response: The limitation is mentioned in the discussion. Differences between fig tea concentrations were also noted in the abstract.
Introduction
While the introduction explains NAFLD pathophysiology in detail, the link between isoschaftoside (ISS) and macrophage polarization is not clearly established.
Response: An extremely important point: the link between ISS and macrophage polarization was suggested for the first time in this study and had not been found before. On the other hand, apigenin, an ISS aglycon, has been reported to regulate macrophage polarization. This information is therefore included in the Introduction.
More emphasis is needed on why ISS is a particularly promising therapeutic approach compared to existing mechanisms.
Response: I apologise for any misunderstanding; I do not believe that ISS alone is a promising treatment option. I think that a combination of treatments with different mechanisms may be more effective.
The epidemiological context is addressed, but more detailed data on incidence trends in different populations would add value.
Response: Added to Introduction.
Materials and Methods
The methodology for ISS dosing and fig leaf tea preparation is described, but it remains unclear whether the administered doses are directly translatable to humans.
Response: The low concentration fig leaf tea is the concentration used in a previous human intervention study. In that study, it was found to actually improve mild atopic dermatitis, and was used in this study as a concentration that would be expected to be effective in humans. The high concentration fig leaf tea was set as the upper limit concentration for human consumption. The concentration of ISS was 50 µM, which was the lowest level of the fig cultivars investigated in the previous study. Please check the additional description (section 2.1 & 2.2.).
More details on statistical analysis are needed, particularly regarding the control of potential confounders.
Response: Fig leaf tea is a mixed composition and it is difficult to control for potential confounding factors. Future work will include correlating the ameliorative effect with the amount of ISS from samples of ISS-added fig leaf tea.
The sample size (n = 6 per group) is relatively small, which may limit the statistical power of the findings.
Response: In the context of the 3Rs (reduction), six animals per group was chosen as the minimum number to achieve statistical significance in this study. Unfortunately, I also believe that the statistical power of the study was limited by the reduction of arguments that occurred during the course of the study. In the future, the effect of the ISS will be more clearly defined in a reproducibility study.
No discussion on potential dietary variations or microbiome influences that could affect the results.
Response: The absorption of ISS is expected to vary depending on a person's microbiome, which has been added to the discussion. Please check the additional description (Discussion).
Results
While significant reductions in inflammatory markers were observed in the HF_FT_high group, it is unclear why the HF_FT_low group did not show similar effects. A more detailed explanation is needed.
The differences between ISS alone and fig leaf tea are mentioned, but no in-depth analysis is provided on potential antagonistic or synergistic compounds in fig leaf tea.
Response to both comments: In the Discussion, I described the possibility that the furanocoumarin glycosides in fig leaf tea may have NAFLD promoting effects. Please check the additional description (Discussion).
Supplementary data S3 shows that the expression of genes associated with oxidative stress is lower in fig leaf tea than in ISS. This result may be due to the action of the antioxidants rutin and caffe malic acid in fig tea. Please check the supplementary Data S3.
The gene expression data are extensive, but their functional relevance to known NAFLD pathogenesis mechanisms is not sufficiently discussed.
Supplementary Data S3 shows the expression levels of genes associated with NAFLD mechanisms. The result also shows that none of these genes were commonly reduced in HF_FT high and HF_ISS. Please check the supplementary Data S3 and additional description (section 3.2.).
No clarification on whether the observed inhibition of M1 polarization persists long-term.
Response: The inhibition of M1 polarization in ISS was first suggested in this study. Therefore, persistence should be addressed as a future research topic.
Discussion
The study proposes a hypothesis about the role of M1/M2 macrophage polarization in NAFLD but does not compare it with other established mechanisms of NAFLD progression.
Response: Supplementary Data S3 shows the expression levels of genes associated with NAFLD mechanisms. The result also shows that none of these genes were commonly reduced in HF_FT high and HF_ISS. Please check the supplementary Data S3 and additional description (section 3.2.).
The claim that ISS inhibits TLR4-mediated inflammatory responses lacks direct functional experiments (e.g., knockdown or knockout models) to confirm the mechanism.
Response: The only data in this study suggesting suppression of M1 polarisation in the ISS are the results of gene expression analysis. More direct evidence using M1/M2 antibodies should be obtained in the future. The TLR knockout experiments you suggest are very interesting and I would like to do them if I have the opportunity.
While clinical translation is briefly mentioned, potential safety concerns or side effects of long-term ISS supplementation are not addressed.
No discussion on possible sex-specific differences or individual variability in response to ISS or fig leaf tea.
Response to both comments: In a previous human intervention study, low dose fig leaf tea containing 75 µM ISS was consumed for two months with no subsequent blood tests or self reported abnormalities. Therefore, I think that drinking 50 µM ISS for two months, which was found to be effective in this study, would be safe. However, long term consumption of high concentrations of ISS, for example in supplements, needs to be checked. As mentioned above, I think that the absorption of ISS will vary from person to person.
Conclusion
The conclusion is broad and somewhat speculative, implying that ISS has general anti-inflammatory properties without clearly limiting its effects to NAFLD.
More precise suggestions for future research are needed, particularly regarding clinical trials or alternative models to investigate the long-term effects of ISS.
Response: As mentioned above, I have revised the content of the report to limit it to the effects of ISS in NAFLD. I have also included the need to investigate the long-term effects of ISS.
Sincerely,
Tatsuya Abe
Round 2
Reviewer 3 Report
Comments and Suggestions for Authors
None